# Novel Competitive Fluorescence Sensing Platform for L-carnitine Based on Cationic Pillar[5]Arene Modified Gold Nanoparticles

**DOI:** 10.3390/s18113927

**Published:** 2018-11-14

**Authors:** Xiaoping Tan, Yang Yang, Shasha Luo, Zhong Zhang, Wenjie Zeng, Tingying Zhang, Fawu Su, Linzong Zhou

**Affiliations:** 1Key Lab of Inorganic Special Functional Materials, Chongqing Municipal Education Commission, School of Chemistry and Chemical Engineering, Yangtze Normal University, Fuling 408100, China; 15334598631@163.com (Y.Y.); 15215297918@163.com (S.L.); 18225211657@163.com (Z.Z.); 17623010998@163.com (W.Z.); 18875144840@163.com (T.Z.); 2State Key Laboratory for Conservation and Utilization of Bio-Resources in Yunnan, Yunnan Agricultural University, Kunming 650224, China; 3School of Geographical Science and Tourism Management, Chuxiong Normal University, Chuxiong 675000, China

**Keywords:** cationic pillar[5]arenes, host–guest recognition, Au nanoparticles, L-carnitine

## Abstract

Supramolecular host-guest interaction and sensing between cationic pillar[5]arenes (CP5) and L-carnitine were developed by the competitive host-guest recognition for the first time. The fluorescence sensing platform was constructed by CP5 functionalized Au nanoparticles (CP5@Au-NPs) as receptor and probe (rhodamine 123, R123), which shown high sensitivity and selectivity for L-carnitine detection. Due to the negative charge and molecular size properties of L-carnitine, it can be highly captured by the CP5 via electrostatic interactions and hydrophobic interactions. The host-guest mechanism between PP5 and L-carnitine was studied by ^1^H NMR and molecular docking, indicating that more affinity binding force of CP5 with L-carnitine. Therefore, a selective and sensitive fluorescent method was developed. It has a linear response of 0.1–2.0 and 2.0–25.0 μM and a detection limit of 0.067 μM (S/N = 3). The fluorescent sensing platform was also used to detect L-carnitine in human serum and milk samples, which provided potential applications for the detection of drugs abuse and had path for guarding a serious food safety issues.

## 1. Introduction

Gold nanoparticles (Au-NPs) have the characteristics of facile synthesis, high chemical stability and easy surface functionalization, which have attracted the great interest of researchers in the fields of nanotechnology and nanomaterials because of their importance in biomedicine [1] sensing [2] catalysis [3] nanoelectronics [4,5] et al. For this reasons, many novel hybrid nanomaterials are designed to synthesis Au-NPs with morphology, size and compositional control [6,7]. The interactions of nanoparticles with surface ligands are also a key aspect for many applications. For example, appropriate surface modification of nanoparticles can improve their sensor potential.

Macrocyclic arenes play an important role in the field of supramolecular chemistry [8,9,10,11]. After crown ethers, calixarenes, cyclodextrins and cucurbiturils [12,13], pillar[n]arenes are the fifth class of macrocyclic host molecules and they were first reported by Ogoshi [14] Pillar[n]arenes mainly consist of pillar[5]arenes and pillar[6]arenes, which are linked by methylene bridges at their *para*-positions to form a unique rigid pillar architecture. Pillar[n]arenes are important players in supramolecular chemistry due to their easy synthesis, unique pillar shape, symmetrical structure, excellent host-guest properties and natural supramolecular assembly characteristics [15]. They have numerous potential applications in host-guest chemistry [16,17,18] hybrid nanomaterials [19] and biomedical material [20,21,22,23,24]. At present, much research has focused on the synthesis of pillar[n]arenes [25,26] their host−guest chemistry and supramolecular self-assembly. The combination of metal nanoparticles and supramolecular macrocyclic compounds can produce strong synergistic effects and improve the properties of nanoparticles, where host-guest chemistry can play an important role [27]. However, the conjugation of pillar[n]arenes with Au-NPs and the application of the resulting hybrid nanomaterials are still rarely reported [28].

The stabilization of ligands possessing carboxyl (−COOH), sulfhydryl (−SH) and amine (−NH_2_) groups is crucial to the synthesis and stabilization of Au-NPs [29,30,31]. Yang et al. produced a new carboxylatopillar[5]arene-modified Au-NPs with good dispersion and narrow size distributions in aqueous solution, the supramolecular self-assembly of CP[5]A@Au-NPs is very useful for sensor and detection of the paraquat [32]. Huang et al. first reported a novel type of amphiphilic Au-NPs with bilayers of an amphiphilic pillar[5]arene modified on their surfaces, which can be used in the fabrication of self-assembled composite microtubes (SCMTs) and these SCMTs are excellent catalysts [33]. Pastoriza-Santos et al. prepared an ammonium pillar[5]arene-stabilized Au-NPs with shape and size control by using seeded growth. The ammonium pillar[5]arene-stabilized Au-NPs were applied to detect 2-naphthoic acid and polycyclic aromatic hydrocarbons [34]. Recently, Yang et al. fabricated the green synthesis of hydroxylatopillar[5]arene-modified Au-NPs (HP5@Au-NPs). The HP5@Au-NPs can self-assemble into multiple well-defined architectures and employ as not only scaffolds but energy acceptors for turn-on fluorescence sensor based on a competitive host−guest interaction [35]. Despite enormous development has been achieved in self-assembly of pillar[5]arene-based Au-NPs in recent years, it still remains a great need to further develop their application of the resulting hybrid nanomaterials.

L-carnitine, also known as (3R)-3-hydroxy-4-(trimethylammonio)butanoate, is a naturally occurring substance that is essential for fatty acid oxidation and energy production in the human body [36,37]. Deficiency of L-carnitine leads to major energy loss and toxic accumulations of free fatty acids. Though many methods have been used for detecting L-carnitine, such as chromatography [38], capillary electrophoresis [39,40] voltammetric [41] fluorescence [42,43,44] and so forth, there are still great challenges in finding selective sensitive tools to detect L-carnitine. Fluorescence technique as a promising method for detection of L-carnitine has many advantages over other common detection techniques, such as good portability, low-cost, high sensitivity and selectivity [45].

Herein, we describe a simple and convenient “turn-off-on” fluorescent sensing platform between the cationic pillar[5]arenes (CP5) and L-carnitine. The fluorescence sensing platform is constructed by CP5@Au-NPs as receptor and probe R123, which has high sensitivity and selectivity to detect L-carnitine. A competitive fluorescence sensing platform based on CP5@Au-NPs is illustrated in Scheme 1. This method is simple, low cost, sensitive and selective and has been applied to L-carnitine detection in human serum and milk samples.

## 2. Experiment Section

### 2.1. Reagents and Apparatus

HAuCl_4_, Rhodamine 123 (R123), L-carnitine and NaBH_4_ were obtained from Shanghai Titan Scientific Co. Ltd (Shanghai, China). CP5 was synthesized by the literature [46,47] and the synthetic route is shown in Appendix A. The structure and purity of all compounds were confirmed by ^1^H NMR and ^13^C NMR (see Appendix A). Detailed synthetic procedures and analytical data are given in the Supporting Information (SI). Other chemicals were of analytical grade. Deionized water (DW, 18 MΩ cm) was used to prepare all of the aqueous solutions.

### 2.2. Apparatus and Instruments

The samples were characterized by Fourier transform infrared (FTIR) spectroscopy via the SCIENTIFIC Nicolet IS10 (MA, New York, USA) FTIR impact 410 spectrophotometer using KBr pellets at a wavelength of 4000–400 cm^−1^. The X-ray photoelectron spectroscopy (XPS) was performed on an ESCALAB-MKII spectrometer (VG Co., London, United Kingdom) with Al Ka X-ray radiation as the X-ray source for excitation. EDS was carried out in the JEM 2100 transmission electron microscopy (TEM, Tokyo, Japan) equipped with an energy dispersive X-ray spectrometry. The zeta potential of the sample was measured with a Malvern Zetasizer Nano series. Fluorescent titrimetric experiments were performed on a Hitachi F-4500 spectrophotometer (Tokyo, Japan). ^1^H NMR and ^13^C NMR spectra were recorded on a Bruker Avance DMX-400 spectrometer at 400 MHz and 500 MHz. The deionized water was obtained by the EU-K1-40TJ (Nanjing, China).

### 2.3. Synthesis of the CP5@Au-NPs

The CP5@Au-NPs composite was obtained based on similar work [28,34]. The CP5@Au-NPs was synthesized by reducing HAuCl_4_ in presence of CP5. In a typical synthetic procedure, an aqueous solution of CP5 (100 μM, 2000 μL) and an aqueous solution of HAuCl_4_ (10 mM, 200 μL) were added to DW (10 mL), then the fresh aqueous solution of NaBH_4_ (40 μL, 0.1 M) was added into the mixture under vigorous stirring. And the solution got wine red, which indicated that cationic pillar[5]arene-modified Au nanoparticles were prepared.

### 2.4. Experiments for Titration L-carnitine

Aqueous solutions of R123 (200 μM), L-carnitine (400 μM) and CP5@Au-NPs (1.0 mg mL^−1^) were prepared. A final concentration of 2 μM R123 was also obtained via dilution. By gradually addition of CP5@Au-NPs dispersion to the R123 solution and the fluorescence of the R123 was gradually quenched. The competitive displacement experiments were performed as follows: the L-carnitine solution was gradually added to a complex of R123-bound CP5@Au-NPs to displace the R123 molecule from the cavity of CP5 by L-carnitine. The fluorescence signal was measured and recorded after the combined solution was mixed by vortexing for 3 min.

## 3. Results and Discussion

### 3.1. Characterization of the CP5@Au-NPs

Firstly, the UV-vis absorption of [CP5]/[HAuCl_4_] at different concentrations was obtained according to the surface plasmon resonance (SPR) of Au-NPs at ~520 nm, as shown in Appendix A. The SPR peak maximum was almost unchanged when the value of [CP5]/[HAuCl_4_] was increased from 0.05 to 1, which suggested that the excess CP5 had little influence on the sizes of Au-NPs. The synthesized CP5@Au-NPs was wine red (Appendix A) and the UV-vis spectroscopy absorption was shown at ~520 nm (Figure 1A), which demonstrated that CP5 stabilized and modified Au-NPs were successfully synthesized. We further studied the morphology features of CP5@Au-NPs by TEM. As shown in Figure 1B, CP5 modified Au nanoparticles (CP5@Au-NPs) with spherical structure were successfully prepared and the size of CP5@Au-NPs was greatly uniform and homogeneous dispersion, which was ascribed to the outstanding size regulating and stabilized effect of CP5 by the coordination between Au-NPs and quaternary ammonium salt groups of CP5. The high resolution transmission electron microscopy (HRTEM) images of CP5@Au-NPs (Figure 1C,D) shown that the corresponding CP5@Au-NPs diameter and crystal lattice spacing were approximately 7 nm and 0.285 nm, respectively. The small size and homogeneous dispersion of CP5@Au-NPs lead to the high catalytic activity and fluorescence quenching property, indicating that CP5@Au-NPs have potential applications in sensing and catalysis.

The CP5@Au-NPs were characterized by zeta potential and the results were shown in Figure 2A,B. The *ξ*-potential value of Au-NPs is almost zero mV in general because the surface of Au-NPs has no charge. Therefore, different charge groups modified Au-NPs will lead to different *ξ*-potential value. By comparing the zeta potential value 12.5 mV of CP5 (Figure 2B), the zeta potential value of CP5@Au-NPs (Figure 2A) are 35.2 mV that is almost three times than that of CP5, which suggests that CP5 successfully modified Au-NPs to form CP5@Au-NPs. FTIR spectroscopy was used to verify whether the macrocyclic molecule CP5 was capped on the Au-NPs. From the FTIR spectroscopy of CP5@Au-NPs and CP5 (Figure 2C), as compared to bare CP5, typical absorption peaks at 1625, 1491 and 1407 cm^−1^ of the benzene ring in CP5 and absorption peaks at 2975 and 2846 cm^−1^ of –CH_3_ and –CH_2_− in CP5 were observed, which shown that the Au-NPs were capped by CP5. We used XPS for proving the presence of CP5 on the surface of Au-NPs. As shown in Figure 2D, for pure CP5, three peaks were observed at 532.5, 402.1 and 285.1 of O 1s, N 1s and C1s, respectively. However, for the CP5@Au-NPs, a pronounced Au 4f peak was observed, which further indicated that the Au-NPs had successfully been modified by CP5. As shown in Figure 2E, the Au 4f_5/2_ peak at 87.6 eV and Au 4f_7/2_ peak at 83.8 eV were observed and the results were similar with those reported [48,49,50], which illustrated that CP5@Au-NPs successful were obtained. The prepared CP5@Au-NPs were further characterized by EDS to investigate the elements of CP5@Au-NPs and shown in Figure 2F. Obvious Au elements were observed at CP5@Au-NPs, which further indicated that the modified processes had been taken place between CP5 and Au-NPs. Therefore, the above results could suggest that CP5 had successfully grafted onto the Au-NPs and form CP5@Au-NPs hybrid nanomaterials.

### 3.2. Fluorescence Spectra Analysis

The fluorescence quenching performance with CP5 and CP5@Au-NPs towards R123 were developed and shown in Figure 3A. As we can distinctly see from Figure 3A, the fluorescence intensity of R123 was quenched by CP5@Au-NPs for the reason of fluorescence resonance energy transfer between R123 and Au-NPs. R123 was connected by CP5 to Au-NPs and the fluorescence was quenched. Therefore, the fluorescence intensity of R123 was continuously quenched with the increasing of the CP5@Au-NPs amount and shown in Figure 3B. The probe R123 can enter the cavity of CP5 via the host-guest recognition because of the suitable structure/size, which leads to fluorescence quenching by effective energy transfer from the probe to the Au-NPs. Figure 3C shows that the successive reversion of the fluorescence signal of R123 as the successive increase of L-carnitine in the pre-formed R123@CP5@Au-NPs inclusion complex. The fluorescence signal reversion was caused by the adding of the amount of L-carnitine, which suggested that the successful detection of L-carnitine by this fluorescence approach. Therefore, it can be concluded that R123 entered into the cavity of CP5 and formed an inclusion complex with CP5@Au-NPs. In addition, the R123 molecule was released from the cavity of CP5 by the addition of L-carnitine based on the competitive supramolecular recognition. Herein, the phenomenon of the “turn−off−on” fluorescence process was developed. Besides, R123 was incubated with CP5@Au-NPs to form R123@CP5@Au-NPs composite and attached the Au-NPs and accompanied by the phenomena of indicator fluorescence ‘turn off’ on account of the fluorescence resonance energy transfer (FRET) [35,51,52,53]. Some control experiments had been performed to confirm that the observed fluorescence intensity recovery was caused by the displacement of R123 by L-carnitine from the cavity of host molecule CP5. We further researched the fluorescence reversion of R123 in the presence of Au-NPs. As shown in Appendix A, although the fluorescence quenching phenomenon obviously was apparent between Au-NPs and R123, the fluorescence reversion did not occur with the addition of L-carnitine. Therefore, the process could conclude that the dye indicator R123 was firstly combined with CP5@Au-NPs and then released from CP5@Au-NPs upon the addition of L-carnitine to form a fluorescence “switch-off-on”.

Figure 3D shown a calibration curve for the quantitative determination of L-carnitine and the fluorescence ratio F/F_0_ was proportional to the concentration of L-carnitine. The linear response ranges for L-carnitine were 0.1–2.0 and 2.0–25.0 μM. The detection limit was 0.067 μM (S/N = 3) and the corresponding regression equations of F/F_0_ = 0.41 C (µM) + 1.13 and F/F_0_ = 0.07 C (µM) + 1.48 with correlation coefficients of 0.925 and 0.995 were obtained. This approach was compared to other methods for detection L-carnitine (Appendix A). This competitive fluorescent method exhibited a wider linear range, lower detection limit and high selectivity than previously reported approaches. Moreover, this method is very convenient and simple for the determination of L-carnitine and has potential applications in the detection of L-carnitine in human blood and food.

### 3.3. The Analysis of Host–Guest Recognition

The recognition process between CP5 and L-carnitine was studied by the molecular docking. The inclusion model of L-carnitine with CP5 was obtained by using the AutoDock 4.2.6 [54] and illustrated in Figure 4A,B. Due to the cavity size of CP5 and the molecular size of L-carnitine, they could be recognized by a 1:1 guest–host composite. As we can see from the Figure 4A,B, the negative charge of carboxylate in L-carnitine could form a higher capacity electrostatic interaction with positive charge groups of –N(CH_3_)_3_^+^ in CP5 (Figure 4C). In addition, the quaternary ammonium salt of L-carnitine could form the cation-π interaction with the five benzene rings of CP5 and the quaternary ammonium salt of L-carnitine entered the cavity of CP5 by hydrophobic interaction (Figure 4D). However, it was not insufficiency to explain the recognized mechanism by molecular docking. The host-guest recognition between CP5 and L-carnitine was also studied by the ^1^H NMR and the result was shown in Figure 5. It is clear that the proton Ha and Hb of L-carnitine disappeared after complexation and the H1 and H5 of CP5 have shifted upfield, suggesting that CP5 can bind L-carnitine with more affinity to release R123 and other interference. Therefore, a selective platform of detection L-carnitine by CP5@Au-NPs was obtained. The more affinity electrostatic interaction has taken place between L-carnitine and CP5 due to the negative charge of L-carnitine and the positive charge of CP5, which plays an important role in host-guest interaction [17,55,56,57]. And the guest L-carnitine can be recognized by the host CP5 via electrostatic interactions and hydrophobic interactions.

### 3.4. Selectivity and Practical Samples Analysis

The interference study for detection of L-carnitine with the R123B-bound CP5@Au-NPs was measured with 100-fold concentrations of L-carnitine analogues (DA, UA and AA) and the structures of these interferences were shown in Appendix A. Besides, the interferences with common interferences (100-fold concentration) including NaCl, KCl, MgSO_4_, glucose, sucrose, BSA and tween 20 were also tested. Figure 6A shown that the fluorescence intensity did not change when these interferences were added to R123@CP5@Au-NPs by comparing L-carnitine in the presence of R123@CP5@Au-NPs. Figure 6B shown a significant fluorescence increase upon the addition of L-carnitine. However, the addition of other competitive interferences did not cause significant fluorescence changes, indicating that these interferences did not cause false-positive signal. To assess the R123@CP5@Au-NPs in practical applications, L-carnitine was detected in human serum and milk samples with a standard addition method (Table 1). The recoveries were 93.5–101.7% and the RSDs obtained by this method were 1.6–4.1%. The accuracy and precision of this proposed approach were satisfactory, which indicated that the method could be applied for the determination of L-carnitine in serum and milk samples.

## 4. Conclusions

In summary, we describe a simple and convenient “turn-off-on” fluorescent sensing platform that uses cationic water-soluble pillar[5]arene modified Au-NPs and dye Rhodamine 123 as energy donor-acceptor pair. The outstanding host–guest recognition capability of CP5 and the excellent quenching performance of Au-NPs make this sensing system suitable for L-carnitine detection in human serum and milk samples. This work demonstrates that the CP5@Au-NPs composite is a good energy acceptor for fluorescence sensing platforms with potential applications in many fields.

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
