# Peer review of "Novel Competitive Fluorescence Sensing Platform for L-carnitine Based on Cationic Pillar[5]Arene Modified Gold Nanoparticles"

_sensors, 2018, doi:10.3390/s18113927_

Reviewer 1 Report

This work deals with the development of novel fluorescence sensing platform for L-carnitine based on gold nanoparticles. Authors reported about “turn-off-on” fluorescent sensing platform using cationic water-soluble pillar[5]arene modified Au-NPs and dye Rhodamine 123 as the energy donor-acceptor pair. The designed sensing platform was used to detect L-carnitine in human serum and milk samples. It provides the potential applications of this novel fluorescent sensing system. The work looks like a short communication and may be published after minor revision.

Notes:

1. I think that the Scheme 1B is not correct shown. How does a spatially bulky rhodamine come into the cavity of pillar[5]arene? The inclusion of rhodamine into the cavity of pillar[5]arene should be confirmed by shifting of signals in NMR spectrum of macrocycle. These changes in 1H NMR spectrum we are not observe. Therefore the Scheme 1B should be redrawn.

2. I think that Scheme 2 can be carried over to Supporting Information, because the synthesis of CP5 is known procedure.

3. Synthesis and potential applications of pillar[5]arenes in host-guest chemistry have been reported in literature. Those very important pioneering studies should be cited and compared with the present systems. Some examples, but not limited to these, are Organic Chemistry Frontiers, 2018, 5(19), 2780-2786; ChemNanoMat, 2018, 4(9), 919-923; New J. Chem., 2017, 41, 1820-1826; Chem.Comm., 2016; Org. Biomol. Chem., 2016, 14, 4233-4238.

4. Which instrument was used to prepare the deionized water? It should be mentioned in Experimental section.

5. Why authors did not discuss about the size of CP5@Au nanoparticles in measurement of their zeta potential. Also PDI (polydispersity index) of these nanoparticles in solution should be mentioned.

6. It would be interesting to evaluate L-carnitine detection quantitatively. 

Author Response

Thank you so much for giving us an opportunity to revise our manuscript (No: sensors-384587 entitled “Novel competitive fluorescence sensing platform for L-carnitine based on cationic pillar[5]arene modified gold nanoparticles). We are grateful for the detailed comments and suggestions provided by each of the reviewers, and we believe that your input has greatly improved our revised manuscript. Changes were marked in yellow in the revised manuscript.

Reviewer 2 Report

Manuscript entitled “Novel competitive fluorescence sensing platform for L-carnitine based on cationic pillar[5]arene modified gold nanoparticles”, by Xiaoping Tan , Yang Yang, Shasha Luo, Zhong Zhang, Wenjie Zeng, Tingying Zhang, Fawu Su and Linzong Zhou, describes how the host-guest interaction between cationic pillar[5]arenes and L-carnitine can be used for molecular recognition and sensing by chemical modification of the surface of gold nanoparticles. The structure of the functionalized nanoparticles was studied by UV-vis and IR spectroscopy, transmission electron microscopy, zeta potential measurements and XPS. Rhodamine was used as fluorescent sensor for L-carnitine detection and the mechanism of the host-guest phenomenon was studied by fluorescence, NMR and molecular docking. The fluorescent sensing platform was also used to detect L-carnitine in human serum and milk samples.

The work described in the manuscript meets originality and novelty, and therefore its publication will be of interest for the scientific community as the results represent an advance in the field of host-guest recognition and sensing. The work is well planned, the results are properly described and discussed, and the conclusions are sound and supported by the data.

On the other hand, the English language should be improved.

When the English language has been revised carefully, the manuscript will be ready for publication.

Author Response

(The authors gave the same response as above.)

Reviewer 3 Report

This work describes the ability to detect L-carnitine in biological relvant materials, this is done using gold nanoparticles host-guest complex, based on replacements reactions with R123. As the L-carnitine binds to the Au np complex, R123 is released and then detected using fluoresence. The concentration of released R123 is correlated to uptake of the L-carnitine and is used as a measure to determine the presence of L-carnitine in samples in both a qualitative, and quantitative manner. The results are well described, and the degree of measurement, and choice of measurement techniques is appropriate for the conclusions drawn. Future work should further consider false-positive results that may occur from competitive reactions based on other binding molecules that are possibly found in living/live samples. Rarely are natural biological samples so pristine in reality compared to laboratory grade materials and mixtures. Nonetheless, it must be noted that the authors did discuss this within the article. However, future work may bring more light to the matter.

Author Response

(The authors gave the same response as above.)
